# Restoration of BDNF, DARPP32, and D2R Expression Following Intravenous Infusion of Human Immature Dental Pulp Stem Cells in Huntington’s Disease 3-NP Rat Model

**DOI:** 10.3390/cells11101664

**Published:** 2022-05-17

**Authors:** Cristiane Valverde Wenceslau, Dener Madeiro de Souza, Nicole Caroline Mambelli-Lisboa, Leandro Hideki Ynoue, Rodrigo Pinheiro Araldi, Joyce Macedo da Silva, Eduardo Pagani, Monica Santoro Haddad, Irina Kerkis

**Affiliations:** 1Cellavita Pesquisas Científicas Ltda., Valinhos 13271-650, SP, Brazil; rodrigo.pinheiro.araldi@gmail.com; 2Genetics Laboratory, Instituto Butantan, São Paulo 05503-900, SP, Brazil; dener.souza@butantan.gov.br (D.M.d.S.); nicolecm.lisboa@gmail.com (N.C.M.-L.); 3Azidus Brasil, Valinhos 13271-130, SP, Brazil; leandro.ynoue@azidusbrasil.com.br (L.H.Y.); joyce.macedo@azidusbrasil.com.br (J.M.d.S.); eduardo.pagani@cellavitabrasil.com.br (E.P.); 4Programa de Pós-graduação em Biologia Estrutural e Funcional, Escola Paulista de Medicina (EPM), Universidade Federal de São Paulo (UNIFESP), São Paulo 04023-062, SP, Brazil; 5Hospital das Clínicas, Faculdade de Medicina, Universidade Estadual de Campinas (UNICAMP), Campinas 13083-872, SP, Brazil; haddad.monica@yahoo.com.br

**Keywords:** huntington disease (HD), brain-derived neurotrophic factor (BDNF), human immature dental pulp stem cells (hIDPSC), 3-nitropropionic acid (3-NP), neurotrophic influences

## Abstract

Huntington’s disease (HD) is a neurodegenerative inherited genetic disorder, which leads to the onset of motor, neuropsychiatric and cognitive disturbances. HD is characterized by the loss of gamma-aminobutyric acid (GABA)ergic medium spiny neurons (MSNs). To date, there is no treatment for HD. Mesenchymal stem cells (MSCs) provide a substantial therapeutic opportunity for the HD treatment. Herein, we investigated the therapeutic potential of human immature dental pulp stem cells (hIDPSC), a special type of MSC originated from the neural crest, for HD treatment. Two different doses of hIDPSC were intravenously administrated in a subacute 3-nitropropionic acid (3NP)-induced rat model. We demonstrated hIDPSC homing in the striatum, cortex and subventricular zone using specific markers for human cells. Thirty days after hIDPSC administration, the cells found in the brain are still express hallmarks of undifferentiated MSC. Immunohistochemistry quantities analysis revealed a significant increase in the number of BDNF, DARPP32 and D2R positive stained cells in the striatum and cortex in the groups that received hIDPSC. The differences were more expressive in animals that received only one administration of hIDPSC. Altogether, these data suggest that the intravenous administration of hIDPSCs can restore the BDNF, DARPP32 and D2R expression, promoting neuroprotection and neurogenesis.

## 1. Introduction

Huntington’s disease (HD, OMIM 143100) is a progressive, fatal, highly penetrant autosomal dominant neurodegenerative genetic disorder that leads to motor, neuropsychiatric, and cognitive dysfunction [1]. HD is caused by the expansion of a cytosine–adenine–guanine trinucleotide (CAG) repeat located in the first exon of the *IT15* gene (*locus* 4p16.3), which encodes the huntingtin protein (Htt) [2]. Generally, unaffected individuals have less than 35 CAG repeats, while affected individuals carry more than 36 CAG repeats [1].

The Htt protein is crucial for developing and maintaining central nervous system homeostasis [1]. This Htt is enriched at synaptic terminals, and controls the axonal transport and vesicle trafficking [1]. This is because the Htt interacts with over 200 different proteins, many of them involved in microtubule-mediated axon trafficking, such as Huntington-associated protein 1 (HAP1), kinesin, dynactin subunit p150, and dynein [1]. However, the expansion of the CAG tract, verified in the mutated huntingtin protein (mHtt), reduces the protein solubility, resulting in intracellular aggregates (inclusions) of mHtt in both neurons and glial cells [1]. These mHtt aggregates promote: (i) mitochondrial dysfunctions, which increase (ii) the reactive oxygen species (ROS) generations. Oxidative stress promoted by mHtt causes (iii) microglial activation, which leads to (iv) neuroinflammation [1]. The mHtt also interacts with HAP1 and dynactin, (v) reducing the transport of the brain-derived neurotrophic factor (BDNF) protein crucial for the survival of striatal neurons [1,3]. Combined, these actions lead a progressive cell death (neurodegeneration) of GABAergic medium spiny neurons (MSP)—the cell that corresponds to 95% of striatal neurons [1]. Typically, the onset age for HD is between 25 and 45 years [4,5,6,7,8,9,10]. 

Although there is no available treatment able to stop, postpone, or reverse the progression of HD [8,11], studies have demonstrated that an increase in brain-derived neurotrophic factor (BDNF) levels is associated with neuroprotection and amelioration of neurological signs in the transgenic mice model for HD [12,13]. However, the BDNF is not transported through the blood–brain barrier (BBB) [14]. For this reason, the therapeutic use of BDNF is clinically limited by complex procedures that involve intraventricular administration or the use of nanocarriers. In this sense, the advanced therapy with mesenchymal stromal cells (MSCs) expressing BDNF emerges as a useful therapeutic approach for the HD treatment. This is because the MSCs can cross the BBB and engraft within the brain [15,16]. Confirming the therapeutic potential of MSCs, Pollock et al. [17] demonstrated that bone marrow MSC genetically modified to overexpress BDNF (MSC/BDNF) induced a significant increase in neurogenesis-like activity and increased life expectancy in R6/2 mice (transgenic mouse model for HD). In addition, MSCs secrete a plethora of biomolecules that can act in several steps of HD pathophysiology, reducing neuroinflammation, programmed cell death, cell toxicity, and improving neurons interconnections, as revisited by us [1,18,19].

In this sense, we developed a novel and disruptive technology to isolate human dental pulp stem cells (hIDPSCs) from deciduous teeth [18,19]. For sharing a transcriptomic signature with neural crest cells, the hIDPSCs naturally express and secrete high levels of BDNF and nestin (neuronal markers), as previously reported by us [18,19,20]. These data make the hIDPSCs a potential candidate for the HD treatment. 

Herein, we investigated the therapeutic potential of hIDPSC for HD treatment. For this, two different cell doses, 1 × 10^6^ (low dose) and 1 × 10^7^ (high dose), of hIDPSCs were administrated though an intravenous route in Wistar rats subjected to acute treatment with 3-nitropropionic acid (3-NP)—an animal model for HD. The cells were administrated as a single dose or three consecutive doses with one-month intervals. Our results confirmed that the 3-NP-induced striatal lesions were a useful model for HD, as previously demonstrated in the literature [21,22]. We also demonstrated the hIDPSCs engrafted within the cortex and striatum of 3-NP rats, providing evidence that these cells are able to cross the BBB. In addition, we observed an increase of BDNF levels in both the cortex and striatum of 3-NP rats treated with hIDPSCs. This result indicates that the hIDPSCs can restore the BDNF expression, conferring neuroprotection. Supporting this evidence, we observed a reduced number of necrotic into the striatum of 3-NP rats treated with hIDPSCs. Quantitative immunohistochemical analyses also demonstrated an increase in MSN markers DARPP32 and D2R in both the cortex and striatum of 3-NP rats. These data reinforce that the hIDPSCs have a neuroprotective effect and suggest that these have a neurogenic potential. However, further investigations are needed to confirm this neurogenic effect. Altogether, our data provide evidence that the hIDPSCs are a promising advance therapeutic product for the HD treatment.

## 2. Materials and Methods

### 2.1. Ethical Aspects

All procedures employed in this study were approved by the Nuclear and Energy Research Institute (IPEN—Instituto de Pesquisas Energetics e Nucleares) of the University of São Paulo (USP), São Paulo, Brazil (process number 140/14). Protocols concerning experimental animals’ maintenance, care and handling are by all current Brazilian legislation and internationally recognized norms and protocols. All staff working with experimental animals was fully accredited as staff researchers/technicians and were adequately trained in the use of animals for experimental scientific purposes by current Brazilian regulations.

### 2.2. Cell Culture 

The hIDPSCs used in this study were isolated from the dental pulp of deciduous teeth of a 6-year-old individual (male), using a novel and disruptive technology developed by Kerkis et al. [18] and protected by the US patent number US20160184366A1. The hIDPSCs were cultivated until the fifth passage in basal medium (Dulbecco’s modified Eagle’s medium (DMEM)/Ham’s F12, supplemented with 15% fetal bovine serum, 100 U/mL penicillin, 100 μg/mL streptomycin, 2 mM L-glutamine and 2 nM nonessential amino acids, all from Gibco, Carlsbad, CA, USA) [23]. The MSC phenotype of these cells was confirmed using the criteria defined by the International Society for Cellular Therapy (ISCT) [24,25]. The hIDPSCs employed in this study are CD105-, CD73- and CD90-positive, and CD45-, CD34-, CD11b- and HLA-DR-negative, as previously described by Kerkis et al. [18]. These cells also express high levels of BDNF, as previously demonstrated [26]. The hIDPSCs employed in this study were produced by the Brazilian facility Cellavita Pesquisas Científicas Ltda. (Valinhos-SP, Brazil) according to the good manufacturing practices (GMP) required by the Brazilian Health Regulatory Agency (ANVISA, RDC 508/21) for advanced therapy products [23]. These cells comprise the active component of the NestaCell® product [23].

### 2.3. Cell Characterization

To further characterize the hIDPSCs, there was performed the immunodetection of human nestin through indirect immunofluorescence and BDNF through immunocytochemistry.

Indirect immunofluorescence: A total of 1 × 10^3^ cells were seeded onto 1-well chamber slides (Nunc^TM^ Lab-Tek^TM^ II Chamber Slide^TM^ System, Thermo Fisher Scientific, Carlsbad, USA, reference code 154453). Cells were fixed in 4% paraformaldehyde (in PBS), permeabilized in 0.1% Triton X-100 in PBS (Sigma-Aldrich, St. Louis, CA, USA), and incubated with 5% bovine serum albumin (BSA, Sigma-Aldrich, St. Louis, CA, USA) for 30 min. Next, the hIDPSCs were incubated overnight at 4 °C with the primary monoclonal antibodies anti-human nestin (Santa Cruz Biotechnology, Santa Cruz, CA, USA, reference sc-23927) at a final dilution of 1:100 in PBS. Next, the cells were washed three times with PBS for 5 min, then they were incubated for 1 h at room temperature with FITC-conjugated goat anti-mouse at a final dilution of 1:500 in PBS. Slides were mounted using the Vectashield mounting medium with 4′,6-diamidino-2-phenylindole (DAPI, Vector Laboratories, Burlingame, CA, USA). The material was analyzed using the Carl Zeiss Axioplan Laser Scanning Microscope (LSM 410, Zeiss, Jena, Germany) or Nikon Eclipse E1000 (Nikon, Natori, Kanagawa, Japan). Digital images were acquired with a CCD camera (Applied Imaging model ER 339) and the documentation system used was Cytovision v. 2.8 (Applied Imaging Corp.—Santa Clara, CA, USA).

Immunocytochemistry: A total of 1 × 10^3^ cells was seeded onto 1-well chamber slides (Nunc^TM^ Lab-Tek^TM^ II Chamber Slide^TM^ System, Thermo Fisher Scientific, Carlsbad, CA, USA, reference code 154453). Cells were fixed in 4% paraformaldehyde (in PBS), permeabilized in 0.1% Triton X-100 in PBS (Sigma-Aldrich, St. Louis, CA, USA), and incubated with 5% bovine serum albumin (BSA, Sigma-Aldrich, St. Louis, CA, USA) for 30 min. Next, the hIDPSCs were incubated overnight at 4 °C with the primary monoclonal antibodies anti-human BDNF (Abcan, Cambridge, UK, reference code ab108319) at a final dilution of 1:100 in PBS. Next, the cells were washed three times with PBS for 5 min and then they were incubated for 2 h at room temperature with goat anti-rabbit IgG-HRP (Santa Cruz, reference code sc-2004) at a dilution of 1:500 in PBS. After the incubation, the cells were washed three times with PBS for 5 min, then incubated for one minute with 3,3′-diaminobenzidine (DAB, Dako, Hamburg, Germany). The cells were washed three times with PBS, for 5 min. The slides were mounted using Entellan (Merck, Darmstadt, Germany). Material was analyzed using the Nikon Eclipse E1000 (Nikon, Natori, Kanagawa, Japan). Digital images were acquired with a CCD camera (Applied Imaging model ER 339) and the documentation system used was Cytovision v. 2.8 (Applied Imaging Corp.—Santa Clara, CA, USA).

### 2.4. 3-NP Rat Model Obtainig and Experimental Design 

For this study, a total of sixty (n = 60) male Wistar rats aged 8 weeks and weighing 350–450 g was used. The animals were obtained from the Central Bioterium of Butantan Institute, São Paulo-SP, Brazil. The animal room was maintained at a constant temperature of 23 ± 2 °C, 48% humidity and on a 12/12 h light–dark cycle. Food and water were available ad libitum. The animals were kept in the bioterium of the Genetics Laboratory (Butantan Institute) for seven days before the experiments. All animal experiments were performed during the light phase from 9:00 to 16:00 h.

The animal model for HD was obtained using 3-nitropropionic acid (3-NP), a toxin produced by a number of fungal and plant species, which acts as an irreversible inhibitor of mitochondrial succinate dehydrogenase (SDH) [22]. 3-NP has been used to induce cell death associated with mitochondrial dysfunction and neurodegeneration, mimicking the characteristics of HD [24,27,28]. 3-NP was purchased from Sigma-Aldrich (St. Louis, MO, USA) and dissolved in saline, and the pH was adjusted to 7.4 with NaOH. 

The rats were intraperitoneally (IP) treated with a single dose of 20 mg/kg/day of 3-NP for four consecutive days, as previously described by Colle et al. [29]. The animals were randomized and placed into one of the following eight groups (Table 1): Experimental groups—G0 (dedicated to cell engraftment analysis) and G1, which received a single dose of 1 × 10^6^ hIDPSC; G2, which received three doses of 1 × 10^6^ hIDPSC (one per month along three months), receiving a total of 3 × 10^6^ cells; G3, which received a single dose of 1 × 10^7^ cells; and G4, which received three doses of 1 × 10^7^ hIDPSC (one per month along three months), receiving a total of 3 × 10^7^ cells; Control groups CG1 and CG2 received a single (CG1) or three doses of saline (CG2, one per month), since saline is the cell vehicle of administration. An additional control group (CG3), not treated with 3-NP (wild type), was included. 

The first cell/saline administration occurred 24 h after the last (fourth) 3-NP injection, on the fifth day (D5) through an intravenous route (caudal vein). For this procedure, the animals were individually transferred to the anesthetic induction chamber (30 cm × 20 cm × 17 cm). Anesthesia was induced with 5% isoflurane in oxygen (2.4 L/min) for about two minutes and maintained with 2% isoflurane in oxygen (2.4 L/min). A first group (G0) was euthanatized four days after the administration of 1 × 10^6^ cells in order to verify the capability of these cells to engraft within the brain, as an initial proof of concept. Following confirmation of the hIDPSCs’ engraftment within the brain, the animals which received a single dose of hIDPSC (G1 and G3) or saline (CG3) were euthanatized 30 days after the cell/saline infusion, on day 35 (D35). The animals that received three doses of hIDPSC (G4) or saline (CG2) were euthanatized 30 days after the last cell/saline infusion, on day 95 (D95). In order to verify the hIDPSCs engraftment within the brain, groups G0 and G2 were euthanatized four hours after the hIDPSC infusion, on day nine (D9). There was an additional group (G0). The experimental design of this study is shown in Figure 1.

### 2.5. Nissl Staining

3-NP-induced neuronal damages were evaluated by staining with Nissl staining solution, as previously described by Gao et al. [22]. For this analysis, the paraffin-embedded rat brain was sliced into 4-μm sections. The tissue was deparaffinized and hydrated, stained with Nissl staining solution for 10 min at room temperature, then immersed in PBS three times for 5 min. The slides were mounted with Entellan (Merck, Darmstadt, Germany) and analyzed by Axio Imager A1 microscopy (Carl Zeiss, Dresden, Germany) at a magnification of 10× and 20×.

### 2.6. Vybrant-Labeling 

In order to check the graft of the hIDPSCs within the rat brain, the cells were labeled with the non-toxic dye Vybrant CFDA SE Cell Tracer Kit (Invitrogen, Carlsbad, CA, USA), according to the manufacturer’s instructions. The hIDPSCs labeled were intravenous administrated, as described in Table 1. The rat was euthanized and the brain was collected. The brains were embedded into Tissue-Tek Optimal Cutting Temperature (OCT) compound (Qiagen, Hilden, Germany) and sectioned (5 µm) using a cryomicrotome (Cryostat CM1100; Leica, Wetzlar, Germany). Tissue sections were placed on poly-l-lysine-coated slides (Sigma-Aldrich, St. Louis, CA, USA), which were incubated in cold methanol (Sigma-Aldrich, St. Louis, CA, USA) for 15 min to decrease tissue autofluorescence. Afterward, they were washed three times in Tris-buffered saline (TBS, 20 nM Tris and 150 nm NaCl, pH 7.4) containing 0.05% Tween-20 (Sigma-Aldrich, St. Louis, CA, USA). Finally, slides were mounted in an antifade solution (Vectashield mounting medium) and analyzed by fluorescent microscopy (Axio Imager A1; Carl Zeiss, Dresden, Germany) or confocal microscopy (LSM 510 META; Carl Zeiss). An argon ion laser set at 488 nm for FITC (Chemicon, Temecula, CA, USA) and at 536 nm for cyanine 3 (Cy3; Chemicon) excitation was used. The emitted light was filtered with a 505 nm (FITC) and 617 nm (Cy3) long-pass filter in a laser-scanning microscope.

### 2.7. Immunoistochemistry (IHC)

In order to confirm the graft of the hIDPSC in the rat brain, as well as to verify the therapeutic potential of hIDPSCs, the rat brains were subjected to immunohistochemistry using the anti-human nucleus antibody (anti-hNu, clone 235-1, Sigma-Aldrich, St. Louis, USA, reference code MAB1281). For this, the brain samples were sectioned (5 μm), and the tissue sections were deparaffinized using a routine technique. Then, the slides were incubated with ammonia hydroxide (Sigma-Aldrich, St. Louis, CA, USA) for 10 min and washed four times in distilled water for 5 min each. Next, the samples were subjected to antigen retrieval using a pH 6.0 buffer of sodium citrate (Sigma-Aldrich, St. Louis, CA, USA), in a water bath set at 95 °C for 35 min, followed by 20 min at room temperature. After this step, the endogenous peroxidase was blocked with hydrogen peroxide (Sigma-Aldrich, St. Louis, CA, USA) for 15 min. Next, the samples were incubated overnight at 4 °C with primary antibodies diluted in 5% BSA. In this analysis was used the monoclonal anti-human nucleus (hNu) (Millipore Corporation, Billerica, MA, USA, reference code MAB1281) to confirm the graft of hIDPSCs within the brain at a dilution of 1:20. Then, the slides were washed three times with PBS for 5 min each and incubated for 1 h at room temperature with the anti-rabbit IgG secondary antibody at a dilution of 1:500. Afterward, the slides were washed three times in PBS for 5 min. Finally, diaminobenzidine (DAB; Dako Cytomation) was applied to produce brown staining. The stained sections were counterstained with hematoxylin (Sigma-Aldrich, St. Louis, CA, USA) and observed under a light microscope (Axio Observer; Zeiss, Jena, Germany). This method was also employed to verify the MSC phenotype after the cell engraftment and evaluate the expression levels of BDNF, DARPP32 and D2R. For these analyses was used the following primary antibodies: Anti-human BDNF (ab108319), -DARPP32 (ab40801), and -D2R (ab150532)—all from Abcan (Cambridge, UK). These antibodies were used at a dilution of 1:200 in PBS with 5% BSA.

### 2.8. Indirect Immunofluorescence 

In order to verify whether hIDPSC differentiated after engraftment, we assessed the expression of CD73 and CD105 (hallmarkers of MSC) in the brain sections. For this, the brain samples were sectioned (5 μm), and the tissue sections were deparaffinized using a routine technique. Then, the samples were subjected to antigen retrieval using a pH 6.0 buffer of sodium citrate (Sigma-Aldrich, St. Louis, CA, USA), in a water bath set at 95 °C for 35 min, followed by 20 min at room temperature. After this step, the samples were incubated overnight at 4 °C with primary antibodies anti-human CD73 (ab133582) and -CD105 (ab2529), both from Abcan (Cambridge, UK), at a dilution of 1:100 in PBS. After this incubation, the slides were washed three time in PBS for 5 min, then they were incubated for 1 h with the FITC-conjugated goat anti-mouse antibody at a final dilution of 1:500 in PBS. Slides were mounted using the Vectashield mounting medium with 4′,6-diamidino-2-phenylindole (DAPI, Vector Laboratories, Burlingame, CA, USA). Material was analyzed using the Carl Zeiss Axioplan Laser Scanning Microscope (LSM 410, Zeiss, Jena, Germany) or Nikon Eclipse E1000 (Nikon, Natori, Kanagawa, Japan). Digital images were acquired with a CCD camera (Applied Imaging model ER 339) and the documentation system used was Cytovision v. 2.8 (Applied Imaging Corp.—Santa Clara, CA, USA).

### 2.9. Quantitive Analysis Using Image J Software

Transversal sections of striatum and cerebral cortex from three rats (n = 3) of each group: G1-G4 and CG1 and CG2 (Table 1) were employed for the quantitative analysis. For this analysis, six fields of the brain interest areas (striatum and cortex) from each slide were captured using a binocular Nikon Eclipse light microscope (Nikon, Natori, Japan) at the bright field. Images were captured at ×20 magnification using color video camera Nikon CS-R 1(Nikon, Japan) attached to a computer system. Before capturing the images, the light settings were standardized for all imaging sessions. 

The Image J recent version software was downloaded from WCIF Image J (http://www.uhnresearch.ca/facilities/wcif/imagej/installing_imagej.htm (accessed on 28 April 2022). The installed Image J software was opened and the saved image was dragged and inserted in the software. The image was calibrated to the magnification; we clicked on the Plugins/Spatial calibration/Microscope Scale. The objective of the microscope TL × 20 was chosen. Camera Binning was kept in 1 × 1. Then, we clicked on Global Calibration and we gave an okay. The image was adjusted by subtracting the background using the command “Process → subtract background” from the file menu. The digitalized area was submitted to the plug-in “color deconvolution” using the built-in vector HDAB, where the staining of hematoxylin and diaminobenzidine (DAB) was separated into 3 different panels, with hematoxylin, DAB-only image and background. After closing the hematoxylin and the background screen, we clicked on the Image/Adjust/Threshold. Threshold 0 refers to the darker tones of DAB and 255 to the lighter tones. We selected the tonal variation range threshold for each sample. To quantify the selection, we clicked on Analyze/Analyze Particles, and the software gave the percentage of the Area fraction based on a variation range of the threshold chosen by each sample.

### 2.10. Statistical Analysis

Following confirmation that the data from IHC quantification are normally distributed (using the Shapiro–Wilks test), the statistical analyses were performed by ANOVA two-way, followed by the Bonferroni’s post hoc test (both with a significance level of 5%). Analyses were performed using the GraphPad Prism v. 5.02 software (GraphPad Softwares, San Diego, CA, USA)

## 3. Results

### 3.1. Expression of Nestin and BDNF by hIDPSCs

Firstly, we verified the expression of nestin (a neural crest stem cell marker [30]) and BDNF in hIDPSCs, since these proteins are closely related to the HD physiopathology [27]. Results confirmed the expression of these markers in hIDPSCs after cryopreservation, thawing, and before hIDPSC transplantation in the 3-NP rat model (Figure 2).

### 3.2. 3-NP Promoted Striatal Damages

In order to demonstrate that the intraperitoneal administration of 3-NP caused striatal damages, the paraffin-embedded brain samples were sectioned, deparaffined, hydrated and subjected to Nissl staining. Results confirmed the neuronal integrity in the striatum of rats was not subjected to the acute treatment with 3-NP (CG3—wild-type, Figure 3A), as expected. However, we verified an expressive neuronal death within the striatum of 3-NP rats (Figure 3B), confirming that the 3-NP caused striatal damages. Similar results were also verified into the striatum of 3-NP rats treated with saline (CG1 or CG2—placebo, Figure 3C). By contrast, we observed a reduced number of apoptotic/necrotic neurons within the striatum of 3-NP rats treated with hIDPSCs (Figure 3D). This result suggests that the hIDPSCs have a neuroprotective potential. 

### 3.3. hIDPSC Homing in the HD Striatum, Cortex, and Subventricular Zone

We analyzed the capacity of the cells to pass through the blood–brain barrier (BBB) and to engraft into target brain areas. Confocal microscopy images demonstrate the presence of hIDPSC stained with Vybrant (green) within the striatum (Figure 4A). The cells show association with capillary and localization in the parenchyma, acquiring different morphologies of pericytes-like and neuron-like cells. Interestingly, two pericytes-like cells were localized close to the capillary (Figure 4A). We also observe the embranchment of axons in the neuron-like cell (Figure 4A). Note those neuron nuclei are light with the nucleolus, different from pericytes-like cells nuclei, which are strongly stained.

Using the anti-human nuclear antigen, we confirm that the cells homed on the striatum (Figure 4B(b1–b3)), in the subventricular zone (SVZ) (Figure 4B(b4)) and the cortex (Figure 4C(c1–c3)). We also observed a hIDPSC graft in the striatal capillaries and striatal parenchyma (Figure 4B(b2)). No staining was observed in the secondary antibody control (Figure 4C(c4)), confirming the specificity of the anti-human nucleus primary antibody.

Additionally, we verified whether hIDPSC differentiated after homing. Thirty days after hIDPSC administration, the cells, which present fibroblast-like morphology, were positive for anti-CD105 (Figure 4D(d1,d2)) and anti-CD73 (Figure 4E(e1,e2)) antibodies suggesting that some human cells were still undifferentiated at that time.

### 3.4. Qualitative Immunohistochemical Staining Assessment for BDNF, DARPP32 and D2R 

In order to evaluate whether the hIDPSCs could increase the expression levels of BDNF, firstly, we analyzed the expression of this neurotrophin in three different brain areas (cortex, striatum and caudate nucleus) of rats treated and not treated with 3-NP. This analysis was performed 30 days after the cell infusion (D35). Results showed a strong BDNF expression in the striatum of untreated animals (CG3—Control, Figure 5A), while 3-NP-treated and 3-NP + placebo groups almost lost BDNF expression (Figure 5B,D). In turn, 3-NP + hIDPSC animals’ striatum cells were positively stained for BDNF (Figure 5C), similar to that observed in control animals (Figure 5A), which were neither treated with 3-NP nor received the hIDPSC IV infusion. BDNF expression pattern in the cortex (Figure 5E–H) was similar to that in the striatum (Figure 5A–D): control and 3-NP + hIDPSC groups were strongly positive for BDNF (Figure 5E and Figure 3G). By contrast, 3-NP and 3-NP + placebo groups were negative for this marker (Figure 5F,H). In the caudate nucleus, in the normal animal’s group, BDNF immunostaining was strong (Figure 5I), while in the 3-NP + hIDPSC group it was less intensive (Figure 5K). Both 3-NP-treated and 3-NP + placebo groups revealed weak immunolabeling (Figure 5J,L). 

HD pathology is marked by extensive loss of MSN that shows high DARPP32 expression, a fundamental component of the dopamine-signaling cascade. Therefore, DARPP32 can serve as a marker of neuronal loss. In this sense, we verified the DARPP32 expression in the striatum and cortex of both 3-NP + placebo (Figure 6A,B) and 3-NP + hIDPSC (Figure 6B,C,E,G) and CG3 animals (Figure 6D,H) groups, respectively. Strong immunolabeling is observed in the striatum of 3-NP + hIDPSC (Figure 6B,C) and CG3 animals (Figure 6D). In contrast, DARPP32 immunoreactivity in the striatum (Figure 6A) and cortex (Figure 6B) of the 3-NP + placebo group is weak. Our data suggest possible regeneration of endogenous MSN demonstrated through the intensive expression of the endogenous anti-DARPP32 antibody, following hIDPSC transplantation. MSN is also known to express D2 dopamine receptors (D2R).

The expression of D2 depends on the intracellular calcium-signaling pathway. The last activated through the dopamine D1-D2 receptor heteromer resulting in CaMKIIα activation and BDNF production in striatal postnatal neurons, thus leading to enhanced dendritic branching. Therefore, we also evaluated D2R expression in the striatum of experimental animals in 3-NP + placebo (Figure 7A), 3-NP + hIDPSC (Figure 7B,C), and control animals (Figure 7B) groups. The expression of D2R was not observed in the striatum of 3-NP + placebo animals (Figure 7A). In the striatum of the 3-NP + hIDPSC group, a small number of endogenous neurons expressing D2R (Figure 7B,C) was already observed. However, it was lower than in the striatum of control animals (Figure 7B). This very positive signal indicates modest recuperation of D2R expressing MSN in the striatum. Previously, both receptors’ expression was shown in the rat frontal cortex [28]. Thus, we also observed the induction of this expression in the cortex of 3-NP + hIDPSC and normal groups of animals.

### 3.5. Quantitative Immunohistochemical Staining Assessment for BDNF, DARPP32 and D2R

Statistical analysis, based on the number of immunolabelled cells, proved that the treatment with hIDPSCs restored the expression of BNDF, DARPP32 and D2R in the cortex and striatum of 3-NP rats (Figure 8). In the cortex, we observed that the hIDPSCs increased the number of BDNF-positive cells (G1–G4), when compared to the group of animals treated with placebo (CG1) (Figure 8). However, there is no verified statistical differences between the low (1 × 10^6^ cells) and high cell dose (1 × 10^7^ cells, Figure 7). By contrast, we observed an increase of D2R-positive cells within the cortex of 3-NP rats treated with the high cell dose (Figure 8). These results were observed in 3-NP rats treated with both single or triple doses of hIDPC (Figure 8), demonstrating that only one single hIDPSC intravenous administration can restore the expression of these biomarkers in the rat brain. By contrast, for the striatum (the main affected brain area by HD), only the low cell dose increased the number of BDNF, DARPP32 and D2R-positive cells (Figure 8). However, there was not observed statistical differences between the single or triple doses (>0.05), reinforcing that only one single hIDPSC intravenous administration can restore the expression of these biomarkers in the rat brain.

## 4. Discussion

Cumulative evidence has demonstrated that BDNF-expressing MSCs can confer neuroprotection, promoting functional recovery in rodent models of Huntington’s disease (HD) [1,3,17,31,32]. The data make these cells a useful candidate for the treatment of HD, since to date there is no effective therapy able to stop, postpone or reverse the progression of the disease [8,11].

Based on this, we investigated the capability of BDNF-expressing hIDPSCs to restore the BDNF, DARPP32 and D2R expression (proteins whose expression levels are reduced as the neurodegeneration progresses). For this, we subjected Wistar rats to a subacute treatment with 3-NP (20 mg/Kg/day for four days), a toxin that irreversibly inhibits the mitochondrial succinate dehydrogenase (SDH), leading to mitochondrial dysfunction and neurodegeneration of striatal cells, mimicking the characteristics of HD [24,27,28].

Although it is impossible to fully mimic the HD pathology, since no genetically modified nor chemically induced animal models exhibit the main clinical signal of the disease, the chorea [1], the 3-NP has been successfully used to promote selective brain damage that inevitably leads to striatal neurons (MSN) loss [23,33,34]. Furthermore, 3-NP’s mechanism of action occurs through the functional blockade of complex II of the respiratory chain, which reduces the mitochondrial membrane potential and calcium homeostasis. These events lead to increased oxidative stress and, ultimately, the death of striatal neurons in a similar way to that seen in HD [31,32,35,36,37]. For these reasons, the 3-NP has been extensively employed to mimic the HD neurobiological symptoms [23,33,34].

Following confirmation that 3-NP caused striatal damages, which were confirmed by the high number of necrotic neurons verified by the Nissl staining, the animals were intravenously treated with the hIDPSCs labelled with the Vybrant probe. For days after the cell infusion, we showed that hIDPSC reached target brain tissues (cortex, striatum and subventricular zone). These cells were frequently localized tightly with striatal capillaries and in striatal parenchyma. The presence of the hIDPSCs in these areas was also confirmed by the immunodetection of human nuclei, using the human anti-nuclei antibody. These data confirm that the hIDPSCs cross the BBB, as previously reported in studies based on other MSCs [15,16]. 

Interestingly, engrafted cells remained expressing hallmarks of undifferentiated MSC four days after administration. These data indicate that, although the hIDPSCs retain a remarkable in vitro potential for neuronal differentiation, they do not differentiate in vivo to neurons. 

In addition, immunohistochemical staining for BDNF, DARPP32 and D2R antibodies demonstrated the elevated expression of these markers in striatum and cortex in experimental groups (G1-G4), when compared with the control (CG1 and CG2). 

Although the 3-NP animal model does not express the mHtt that, by interacting with HAP1 and dynactin, impairs the BDNF transport along the cortico-striatal pathway [1], we demonstrated a statistically significant reduction in the number of BDNF-, DARPP32- and D2R-positive cells in both cortex and striatum of 3-NP rats, confirming the brain damages produced by the toxin. However, we observed that the treatment with a low (1 × 10^6^ cells) or high dose (1 × 10^7^ cells) of hIDPSCs restored the number of BDNF- and D2R-positive cells to values statistically similar to those verified in rats non-treated with 3-NP (wild-type). By the contrast, we observed that only the higher dose of hIDPSCs was able to increase the number of DARPP32-positive cells into the cortex. However, no statistical difference was observed between the single or three-cell administration. In turns, the analysis of the striatum showed that only the low cell dose increased the number of BDNF-, DARPP32- and D2R-positive cells. Confirming the results observed for the cortex, we did not verify statistical differences between the single or three consecutive administrations of hIDPSCs. These results are in accordance with the literature, which has shown that low cell doses lead to a better outcome [33,34,38]. Detailed analysis of tested markers expression suggests that the dose of 1 × 10^6^ hIDPSCs is enough to confer neuroprotection though the restoration of BDNF expression. This is because the BDNF secretion is required for the cortico-striatal neurons’ survival, and for the regular expression of DARPP32, which is a marker of differentiated striatal MSN and is indispensable for the dopamine-signaling cascade [39,40,41,42,43,44,45]. In this sense, we demonstrated that the hIDPSCs express neuronal proteins, including nestin and BDNF. Thus, the increased expression of BDNF verified in G2–G4 groups confirms that cells engrafted within the brain can restore the BDNF expression, conferring a neuroprotective effect.

In HD pathology, MSN with high expression of DARPP32 is extensively lost [46,47]. Most striatal functions mediated by the MSN, which comprise 95% of the striatum and the rest, are interneurons. MSN expresses the excitatory D1 receptor (D1R) and the inhibitory D2 (D2R) dopamine receptor [46]. The enrichment of D2R in the matrix is protective against dopamine excitotoxicity, providing a neuroprotective effect [48]. Thus, the DARPP32 and D2R increase verified in G2–G4 groups not only provides evidence of the neuroprotective effect of hIDPSCs, but also suggests that the cells can induce neurogenesis through a BDNF-dependent manner. Supporting this hypothesis, studies already demonstrated the upregulation of BDNF [49,50].

The concept of MSC has changed over the years. It is generally accepted that they make bioactive, immunomodulatory and trophic (regenerative) therapeutic molecules. Trophic mechanisms of MSC can reduce chronic inflammation, inhibit apoptosis and scar formation, and stimulate mitosis of tissue-intrinsic progenitors, thus remodeling the damaged tissue [51,52]. BDNF secreting cells are indicative for treating neurodegenerative diseases, especially for HD [53]. Reduction in BDNF expression jeopardizes the long-term survival and morphology of striatal medium spiny neurons (MSN), causing striatal degeneration [29,54,55]. Since neuronal death is a significant cause of HD symptoms, BDNF is the potential not only to maintain neuronal health (neuroprotection), but also to stimulate the growth of new neurons (neurogenesis), which has placed BDNF and BDNF-secreting cells as promising options for HD treatment [12].

Based on these data, we investigated the safety of a low dose of hIDPSCs in patients with HD (phase I clinical trial—available online: https://clinicaltrials.gov/ct2/show/NCT02728115I (accessed on 28 April 2022). The results of this clinical trial demonstrated that the hIDPSCs are well tolerated and improve the HD motor dysfunctions [56], justifying the Phase II clinical trail (available online: https://clinicaltrials.gov/ct2/show/NCT03252535 (accessed on 28 April 2022), which is under analysis.

## 5. Conclusions

Our study shows that intravenous transplantation of BDNF-secreting hIDPSC helps restore the endogenous BDNF expression and the expression of MSN markers (DARPP32 and D2R) in the striatum and cortex of the HD rat model. Loss of MSN is a principal cause of death of HD patients. Therefore, hIDPSC intravenous transplantation is a promising tool for the treatment of HD. However, to confirm this statement, clinical studies are needed.

## Figures and Tables

**Figure 1 cells-11-01664-f001:**
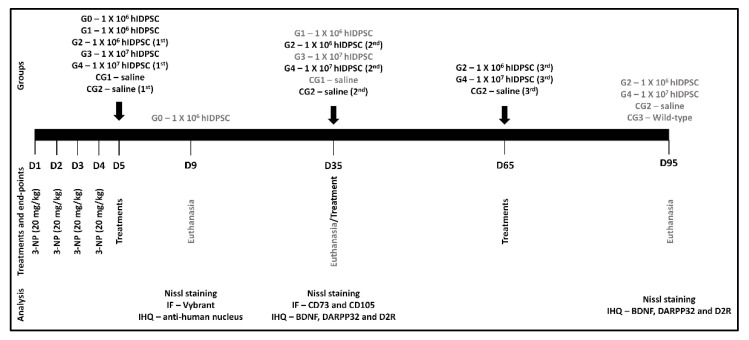
Schematic representation illustrating the study experimental design. 3-NP—3-nitropropionic acid, D—Day, IF—immunofluorescence and IHQ—Immunohistochemistry.

**Figure 2 cells-11-01664-f002:**
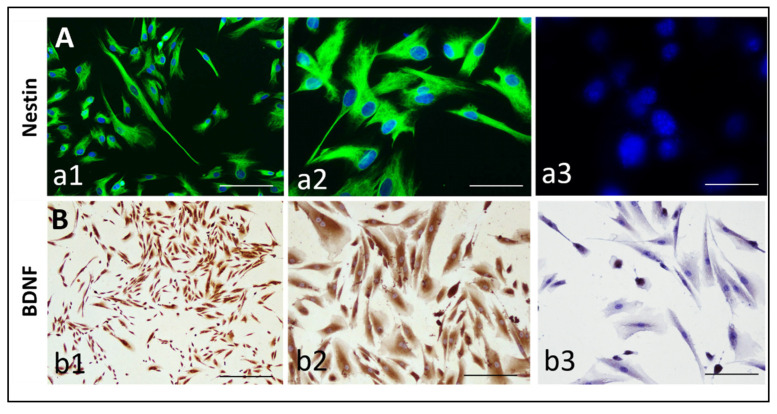
Expression of Nestin (**A**) and BDNF (**B**) in hIDPSC (**A**). Images show the immunodetection of: (**A**) nestin (by indirect immunofluorescence) in two magnifications (20× **a1** and 40× **a2**), as well as its respective negative control, showing the absence of unspecific labeling of secondary antibody (40×, **a3**); and (**B**) BDNF (by immunocytochemistry), in two magnifications (20× **b1** and 40× **b2**), as well as its respective negative control, showing the absence of unspecific labeling of a secondary antibody (40×, **b3**). Scale bars: 25 µm. Cells in fifth passage.

**Figure 3 cells-11-01664-f003:**
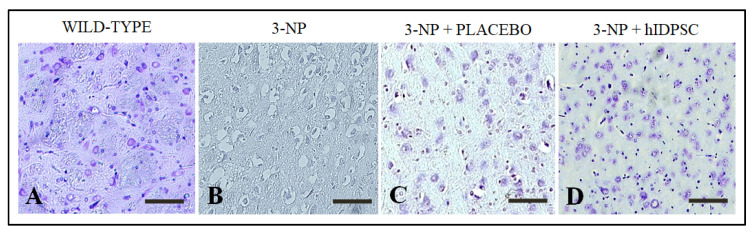
Nissl staining showing: (**A**) Neuronal integrity within the striatum of rats not treated with 3-NP (wild-type), Necrotic neurons within the striatum of rats treated with 3-NP (**B**) and saline (placebo). (**C**). Reduced number of necrotic neurons within the striatum of 3-NP rats treated with hIDPSCs. (**D**). Imagens captures using objectives of 20× (**A**–**C**) and 10×. (**D**). Scale bar of 50 μm (**A**–**C**) and 100 μm. (**D**). Images observed in rats euthanized at day 95 (D95).

**Figure 4 cells-11-01664-f004:**
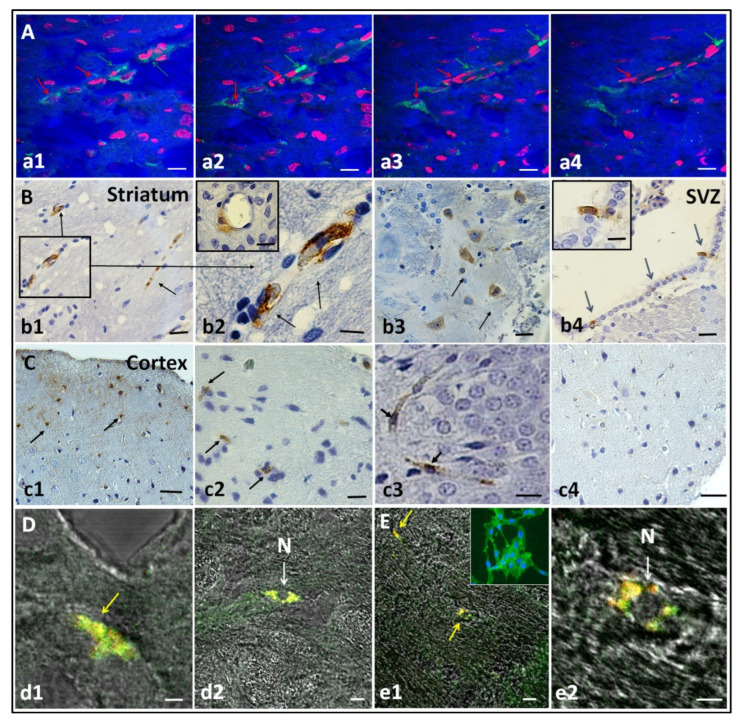
Engraftment of hIDPSCs four days after IV administration in the striatum, cortex and subventricular zone (SVZ). (**A**) Images show hIDPSC (within striatum) stained with Vybrant (green arrows) and nuclei stained with propidium iodide (red arrows). Analysis performed at day 9 (D9). (**B**,**C**) Immunohistochemistry demonstrates positive immunostaining for the anti-hNu antibody (black arrows) observed within the striatum (**b1**–**b3**), SVZ (blue arrows) (**b4**) and cortex (**c1**–**c3**). B2 and B4 show cells immunolabeled with the anti-hNu antibody in the striatal perivascular area and subventricular zone, respectively. Negative control shows the absence of unspecific labeling of secondary antibody control for anti-human nuclei (**c4**). Analysis performed at day 9 (D9). Immunofluorescence assay (in confocal microscopy) showing the expression of CD73 (**D**) and CD105 in the rat brain (**E**). Yellow arrows demonstrate cells expressing CD73-positive (d1) and CD105-positive (e1) and, white arrows, cells C73 (d2) and CD105 in smallest magnification (e2). These cells exhibits an usual MSC phenotype (N). Analysis performed at day 35 (D35). Total magnification: 5× (**a1–a4**,**b1**,**b4**,**c1**,**e1**), 10× (**c4**,**d2**), 20× (**c2**) and 40× (**b2**,**b3**,**c3**,**d1**,**e2**). Scale bars: 5 µm (**a1**–**a4**,**b2**,**b4**,**c2–c4**,**d1**,**d2**,**e1**,**e2**), 10 μm (**b3**).

**Figure 5 cells-11-01664-f005:**
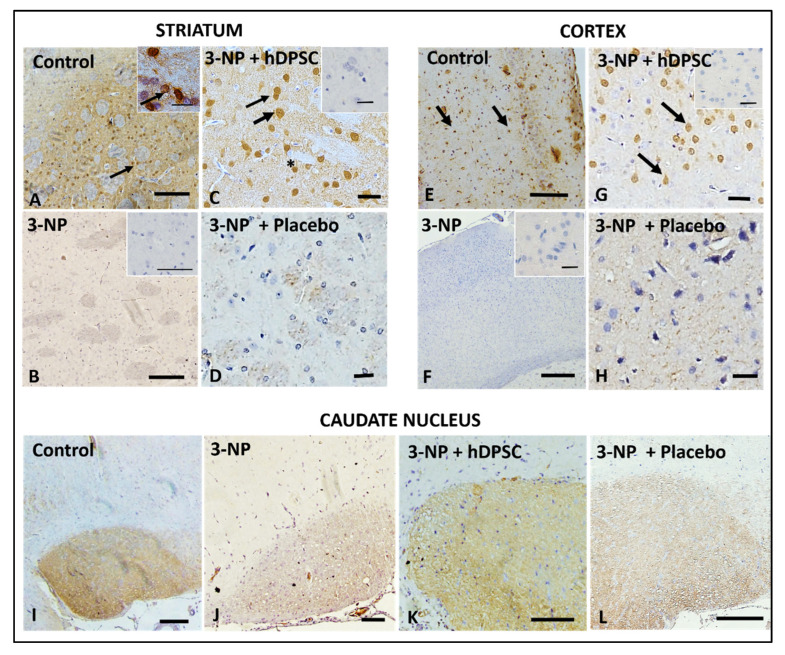
Expression of anti-BDNF antibody (black arrows) in the striatum (**A**–**D**), cortex (**E**–**H**) and caudate nucleus (**I**–**L**) in control (**A**,**E**,**I**); 3-NP (**B**,**F**,**J**), 3-NP + hIDPSC (**C**,**G**,**K**) and 3-NP + placebo animals (**D**,**H**,**L**), respectively. Insets in (**A**–**C**,**F**,**G**) show negative control (secondary antibody). Black arrows indicate BDNF expressing cells. Sections are contra-stained with HE. Scale bars: 50 µm (**A**,**B**,**E**,**F**,**I**–**L**) and 100 µm (**C**,**D**,**G**,**H**). Analysis performed 30 days after (D35) the treatment). Asterisk indicates the enlarged areas. The asterisks indicate the areas that were analyzed in magnification.

**Figure 6 cells-11-01664-f006:**
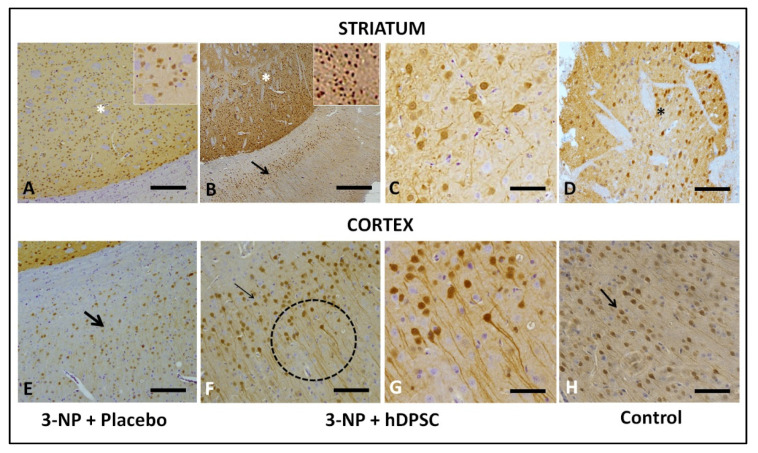
Expression of the anti-DARPP32 antibody in the striatum (**A**–**D**) and cortex (**E**–**H**) in 3-NP + Placebo (**A**,**E**), 3-NP + hIDPSC (**B**,**C**,**F**,**G**) and in control (**D**,**H**) animals, respectively. Black arrows indicate anti-DARPP32 expressing cells. (**C**,**G**) demonstrate high magnification of (**B**,**F**). Sections are contra stained with HE. Scale bars: 50 µm (**A**,**B**,**E**,**F**), 25 µm (**C**,**G**) 10 µm (**D**,**H**). Analysis performed 30 days after the treatments (D35). Asterisk indicates the enlarged areas.

**Figure 7 cells-11-01664-f007:**
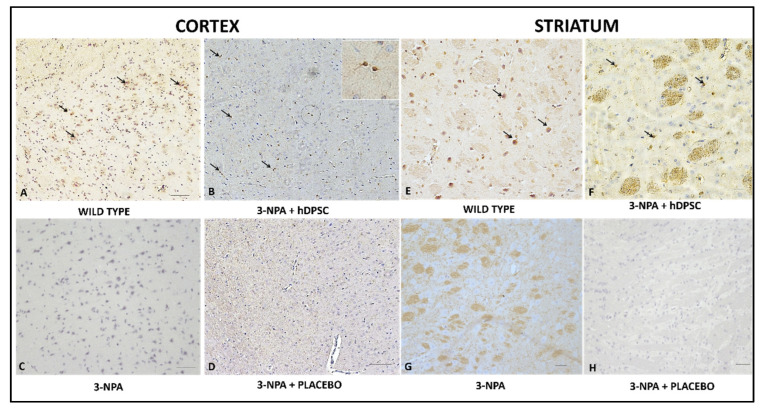
Analysis of anti-D2R antibody expression in the cortex (**A**–**D**) and striatum (**E**–**H**) in four studied groups: wild type (**A**,**E**); 3NPA+ hIDPSC (**B**,**F**); 3NPA (**C**,**G**) and 3NPA + Placebo (**D**,**H**), respectively. Results show positive immunostaining for anti-D2R antibody in the cortex (**A**) and striatum (**E**) of wild-type rats, positive control, as well as in the cortex (**B**) and striatum (**F**) of 3NPA + hIDPSCs. The loss of D2R expression is observed in the cortex (**C**,**D**) and striatum (**G**,**H**) of rats treated with 3NPA, as well as in rats 3NPA + Placebo, respectively. Black arrows indicate anti-D2R expressing cells. Sections are contra stained with HE. Scale bars: (**A**–**D**,**C**,**H**) = 50 µm; (**E**,**F**) and Inset in (**B**) = 10. Analysis performed 30 days (D35) after the treatments.

**Figure 8 cells-11-01664-f008:**
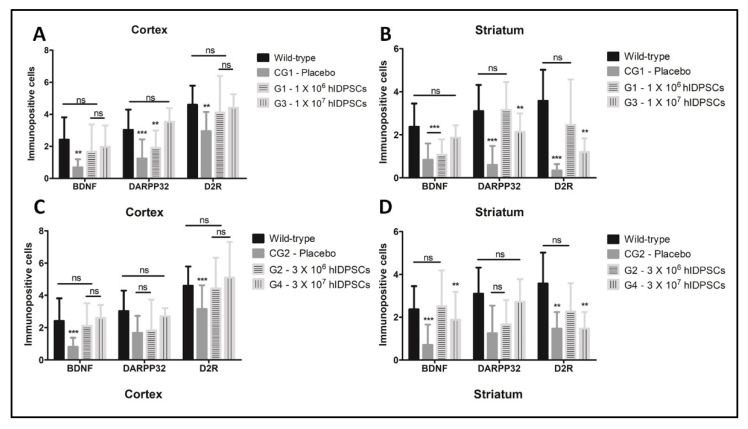
Quantitative analysis showing the number of DARPP32, BDNF and D2R-positive cells in targets regions of striatum and cortex in groups that received a single (**A**,**B**) or three administrations of hIDPSC (**C**,**D**). Results show that the intravenous administration of hIDPSCs increased the expression of these three markers in cortex and striatum (**A**–**D**). In the cortex, results show no statistical difference between the number of BDNF-positive cells in 3-NP rats treated with low and high hIDPSCs (**A**,**C**). However, it is observed that only the high cell dose increased the number of DARPP32- and D2R-positive cells within the cortex. In turn, only the low dose increases the number of BDNF-, DARPP32- and D2R-positive cells into the striatum (**B**,**D**). However, results show the absence of statistical differences between the single and triple doses (*p* > 0.05). Statistical analysis performed using ANOVA two-way, followed by the Bonferroni’s post hoc test, both with significance levels of 5%: ** (*p* < 0.01), *** (*p* < 0.001). *p*-values > 0.05 indicate the absence of statistical differences (ns: non-significative).

**Table 1 cells-11-01664-t001:** Experimental design, describing experimental (G0–G4) and control groups (CG1–CG3), the number of cells administrated and the number of rats used in each assay.

			Number of Rats Used in Each Assay		
			hIDPSC Homing		Immunodetection of
Group	Dose	Total of Cells	Vybrant	Anti-hNu	DARPP32, BDNF and D2R
G0	Single	1 × 10^6^	3	3	N/A
G1	Single	1 × 10^6^	N/A	3	6
G2	Three	3 × 10^6^	3	3	6
G3	Single	1 × 10^7^	N/A	3	6
G4	Three	3 × 10^7^	N/A	3	6
CG1	N/A	N/A	N/A	N/A	6
CG2	N/A	N/A	N/A	N/A	6
CG3	N/A	N/A	N/A	3	6

N/A—Not applied.

## Data Availability

Not applicable.

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
