# Peer review of "Restoration of BDNF, DARPP32, and D2R Expression Following Intravenous Infusion of Human Immature Dental Pulp Stem Cells in Huntington’s Disease 3-NP Rat Model"

_cells, 2022, doi:10.3390/cells11101664_

Round 1

Reviewer 1 Report

The 3-NP rat model which the authors decided to adopt for their analysis has limitations. Even though the authors fail to state this (lines 384-391), however,  they mention the absence of mHTT aggregates and that “mhtt verified in patients with HD leads to loss of DARPP32-expressing medium spiny neurons and D2R expressing neurons of the nigrostriatal pathway. This occurs because the mHtt impairs BDNF…”. I believe this is an aspect that should be elaborated more, as this is crucial for the study.

The increased BDNF expression is beneficial, but it is not clear how? Any improvement in motor, clinical signs, life expectancy?

I am quite surprised that nothing concerning possible mitochondrial restoration have been reported after hIDPSC injection. Mitochondrial dysfunction (and complex II functional blockade) is a direct effect of the 3NP compound in this model.

What is the authors explaination concerning no significant effects in rats receiving three doses of hIDPSC cells?

There is no methodology described for the Anti-hNu (Table 1). The Vybrant tracer was used in G0 for the hIDPSCs homing analysis. But then it is not clear to me why the authors used it again for G2 and not for G1, G3 and G4. Did hIDPSC improve proliferation of endogenous neural stem cells more in the SVZ or this was more or less uniform in all three regions (SVZ, cortex, striatum)?

The authors published an abstract entitled “A PHASE I CLINICAL TRIAL ON INTRAVENOUS ADMINISTRATION OF IMMATURE HUMAN DENTAL PULP STEM CELLS (NESTACELL HDTM) TO HUNTINGTON'S DISEASE PATIENTS”. The authors should comment on this trial, and how this relates to the present study.

Author Response

The 3-NP rat model which the authors decided to adopt for their analysis has limitations. Even though the authors fail to state this (lines 384-391), however,  they mention the absence of mHTT aggregates and that “mhtt verified in patients with HD leads to loss of DARPP32-expressing medium spiny neurons and D2R expressing neurons of the nigrostriatal pathway. This occurs because the mHtt impairs BDNF…”. I believe this is an aspect that should be elaborated more, as this is crucial for the study.

We thank you for this comment. However, we would like to clarify that there is no "gold-standard" animal model to study Huntington's disease (HD), as we discussed in a currently published chapter book dedicated to the physiopathology of HD (Kerkis et al. 2022). This is because no animal model exhibits the chorea's significant signal of disease. Moreover, on the one hand, the genetic (transgenic) animal models for HD express the mutated Huntingtin protein (mHtt); on the other hand, they do not show the loss of medium spiny neurons (MSNs), which is responsible for the motor dysfunctions verified since the beginning of the disease. In contrast, the 3-NP induces striatal damages, promoting the death of MSNs. For this reason, the 3-NP rat model has been extensively used to investigate the HD pathophysiology, as well as a model for preclinical studies, as demonstrated in the studies described below:

Kerkis e al. Advances in cellular and cell-free therapy medicinal products for Huntington disease treatment. In From Pathophysiology to Treatment of Hunting’s Disease, IntechOpen, 2022. DOI 10.5772/intechopen.102539

Samar et al. (2021). Liraglutine impreoves cognition and neuronal function in 3-NP rat model of Huntington’s disease. Frontiers in Pharmacology, 2021. DOI 10.3389/fphar.2021.731483

Protopanaxtriol protects against 3-nitroproprionic acid-induced oxidative stress in a rat model of Huntington’s disease. Acta Pharmacologica Sinica, 2015. DOI 10.1038/aps.2014.107

Thus, the statement that the “mHtt verified in patients with HD leads to loss of DARPP32-expressing medium spiny neurons and D2R expressing neurons of the nigrostriatal pathway” is by the well-established pathophysiology of the disease, as discussed by us in Kerkis er al. (2022). This occurs because the mHtt interacts with HAP1 and dynactin, leading to de-railing of molecular motors from microtubules tracks and cessation of transport” (Kerkis et al., 2022). This action reduces the “survival of striatal neurons” since the BDNF acts as an anti-apoptotic protein, conferring neuroprotection. To clarify the relevance of the 3-NP animal model and provide evidence about the BDNF role in HD, we modified the Introduction and section 2.4. We also added the Nissl staining giving proof of the striatal damages (section 3.3).

The increased BDNF expression is beneficial, but it is not clear how? Any improvement in motor, clinical signs, life expectancy?

The mHtt affects the BDNF transport from cortical to striatal neurons, as discussed in response to the first comment. This action reduced the survival of striatal neurons, cooperating with the neurodegeneration promoted by the mitochondrial accumulation of mHtt. However, as previously demonstrated by us, the hIDPSCs naturally produce and secrete high levels of BDNF (de Almeida et al., 2011). Thus, the increase of BDNF observed in the cortex and striatum of 3-NP rats treated with the cells confirms that the hIDPSCs crossed the brain-blood barrier and engrafted within the brain (as confirmed by the anti-human nuclei immunostaining). Moreover, this result also demonstrates that the intravenous infusion of hIDPSCs improve the striatal neurons' survival, conferring a neuroprotective effect, as confirmed by the increase in expression levels of the medium spiny neurons markers DARPP32 and D2R. These data are present in the discussion (lines 408-413), as demonstrated below:

“BDNF secretion is required for the survival of the cortico-striatal neurons and the regular expression of DARPP32, a marker of differentiated striatal MSN and indispensable for the dopamine-signaling cascade [33–39]. In this sense, we demonstrated that the hIDPSCs express neuronal proteins, including nestin and BDNF. Thus, the increased expression of BDNF verified in G2-G4 groups suggests that cells engrafted within the brain can restore the BDNF expression, conferring a neuroprotective effect”.

It is worth mentioning that the primary goal of this study was to analyze whether the intravenous administration of hIDPSCs could restore the expression levels of these three pathognomonic markers of HD (BDNF, DARPP32, and D2R), providing evidence of the neuroprotective effect of these cells. In this study, we did not include any behavioral analysis that allowed us to verify motor improvement or clinical signs. Moreover, the experimental design of this study was not performed to evaluate life expectance.

de Almeida FM, Marques SA, Ramalho Bdos S, Rodrigues RF, Cadilhe DV, Furtado D, Kerkis I, Pereira LV, Rehen SK, Martinez AM. Human dental pulp cells: a new source of cell therapy in a mouse model of compressive spinal cord injury. J Neurotrauma. 2011 Sep;28(9):1939-49. doi: 10.1089/neu.2010.1317. Epub 2011 Aug 8. PMID: 21609310.

I am quite surprised that nothing concerning possible mitochondrial restoration have been reported after hIDPSC injection. Mitochondrial dysfunction (and complex II functional blockade) is a direct effect of the 3NP compound in this model.

We thank you for this comment. As mentioned by Reviewer 1, the 3-NP promotes mitochondrial dysfunction leading to cell death, as previously discussed by us (Kerkis et al., 2022). However, animal models are not appropriate to demonstrate mitochondrial restoration. For this purpose, in vitro studies based on the co-culture of neurons (obtained by the differentiation of SH-SY5Y cell line) and previously exposed to a low concentration of 3-NP (5mM) with hIDPSCs are most suitable, as demonstrated by Solesio et al. (2013). We inform you that our group has already performed this analysis in this sense. However, this result makes part of another in vitro study and, therefore, it will be published in a novel manuscript dedicated to this issue.

Solesio et al. showed that 3-nitropropionic acid induces autophagy by forming mitochondrial permeability transition pores rather than activating the mitochondrial fission pathway. Brazilian Journal of Pharmacology, 2013. DOI 10.1111/j.1476-5381.2012.01994.x

However, we added the Nissl staining results to demonstrate the neuronal damages promoted by the 3-NP-related mitochondrial dysfunction.

What is the authors explanation concerning no significant effects in rats receiving three doses of hIDPSC cells?

We apologize for this misinterpretation. To clarify the effects of the therapy based on a single or three-dose, we modified Figure 7 (E and D), comparing the expression levels of BDNF, DARPP32, and D2R in the cortex and striatum of 3-NP rats treated with the hIDPSCs. These analyses demonstrated no statistical differences in BDNF, DARPP32, and D2R expression levels between the single or three-cell doses. Based on these results, we designed the phase I and II clinical trials for HD (available on Phase I (SAVE-HD) - https://clinicaltrials.gov/ct2/show/NCT02728115, and Phase II (ADORE-HD) - https://clinicaltrials.gov/ct2/show/NCT03252535. These studies employed three consecutive administrations (one per month) of low cell doses (1 X 106 and 2 X 106 hIDPSCs/Kg). Interestingly, the low dose (1 X 106 cells/Kg) showed the most prominent effect. However, as we reported, both doses were safe (Macedo et al., 2021). Similar results, changing the paradigm that more cells lead to better outcomes, have been reported in the last years (Konkova et al., 2020, Eylert et al., 2021, Wang et al., 2022).

The first hypothesis that the better outcome in the low dose range was provided by Hart et al. (2014). According to this mathematical model, an excessive amount of stem cells generates a lack of nutrients and possible hypoxia, which leads to cell death. To clarify this point, we added these data to the discussion.

Macedo et al. A phase I clinical trial on intravenous administration of immature dental pulp stem cells (NestaCell HDTM) to Huntington’s disease patients. Cytotherapy, 2021. DOI 10.1016/j.jcyt.2021.02.008

Konkova et al. Mesenchymal stem cells early response to low-dose ionizing radiation. Frontiers in Cell Development Biology, 2020. DOI 10.3389/fcell.2020.584497

Eylert et al. Skin regeneration is accelerated by a lower dose of multipotent mesenchymal stromal/stem cells – a paradigm change. Stem Cell Research & Therapy, 2021. DOI 10.1186/s13287-020-02131-6

Wang et al. 2022. A low dose cell therapy system for treating osteoarthritis: In vivo study and in vitro mechanistic investigations. Bioactive Materials, 2022. DOI 10.1016/j.bioactmat.2021.05.029

Hart et al. Paradoxical signaling by a secreted molecule leads to homeostasis of cell levels. Cells, 2014. DOI 10.1016/j.cell.2014.07.033

There is no methodology described for the Anti-hNu (Table 1). The Vybrant tracer was used in G0 for the hIDPSCs homing analysis. But then it is not clear to me why the authors used it again for G2 and not for G1, G3 and G4. Did hIDPSC improve proliferation of endogenous neural stem cells more in the SVZ or this was more or less uniform in all three regions (SVZ, cortex, striatum)?

We thank you for this comment. Table 1 shows that three rats were destined for the biodistribution analysis with the anti-hNu (groups G0-G4). We apologize for this mistake in the description in Table 1. The methodology for the immunodetection using the anti-hNu is shown in detail in section 2.7. This study does not evaluate the proliferation of neuronal stem cells. The main goal of this study was to demonstrate whether the intravenous infusion of hIDPSCs could lead to the engraftment of the cells within the cortex and striatum and restore the expression of BDNF, DARPP32, and D2R. To answer this and other questions, we are performing novel preclinical data to evaluate whether the treatment with these cells can ameliorate the clinical signals of the HD and promote the proliferation and differentiation of neuronal stem cells. The results of this study will be presented in a new publication; however, we have already demonstrated by RNAseq that the hIDPSCs express high levels of different transcriptional factors related to the proliferation and differentiation of neuronal stem cells to medium spiny neurons, such as Sp1, Sp4, and Klf4. The expression of these transcriptional factors was also validated by qRT-PCR in hIDPSCs isolated from three different donors. Our novel preclinical data have demonstrated that the hIDPSCs increase the expression of the Sp1 factor in the subventricular zone, supporting the evidence reported in this study, indicating an increase in the number of DARPP32 and D2R-positive cells.

The authors published an abstract entitled “A PHASE I CLINICAL TRIAL ON INTRAVENOUS ADMINISTRATION OF IMMATURE HUMAN DENTAL PULP STEM CELLS (NESTACELL HDTM) TO HUNTINGTON'S DISEASE PATIENTS”. The authors should comment on this trial, and how this relates to the present study.

We thank you for this comment. As requested, the phase I clinical study, using the hIDPSCs employed in the preclinical study described in our manuscript, was cited as this reference in the final of our discussion, demonstrating the safety and indicating the therapeutic efficacy of these cells for patients with HD.

Reviewer 2 Report

This study describes the improvement of BDNF, DARPP32 and D2R with intravenous administration of human immature dental pulp stem cells into 3-NP rat model. The authors also noted that conditions for transplanted cells were shown (e.g. cell number and transplantation times). This is a rather elaborate study design but which requires some modification. Taking issues as they arise in the paper, I would have the following specific comments:

  1. For the clinical application of cell therapy, the methods for the cell injections should be determined with more consideration. The authors need to discuss more the effects of cell transplantation, especially number of cells at one time (between group 1 and group 3) and administration times (between group 1 and group 2 ,and between group 3 and group 4).
  2. From the experimental design, rat brain tissues were collected at 4 days and 30 days after hIDPSCs administration. What is the reason for 4 day and 30 day -points ?
  3. Page5 line 232-234 and Figure 1B : Expression of BDNF in hIDPSCs were shown by immunocytochemistry. However, the method led to this result is not described. Please check this point.
  4. Page6 line 257-261 and Figure 2D,2E : Expression of CD73 and CD105 in rat brain were shown by immunofluorescence. However, the methods led to these results are not described. Please check these points.
  5. Page8 line 295: Is scale bars of figure3E 100 µm? Please check it.
  6. Page8 line 299-304: In this part, figure number and figure legends description do not match. Please check it.
  7. Page9 line 320-327: In this part, figure number and figure legends description do not match. Please check it.
  8. Page11 line 369: Although it is describe as “1×107cells(G3 and G5)”,G5 is not in Table1. Please check it.

Author Response

This study describes the improvement of BDNF, DARPP32 and D2R with intravenous administration of human immature dental pulp stem cells into 3-NP rat model. The authors also noted that conditions for transplanted cells were shown (e.g. cell number and transplantation times). This is a rather elaborate study design but which requires some modification. Taking issues as they arise in the paper, I would have the following specific comments:

  1. For the clinical application of cell therapy, the methods for the cell injections should be determined with more consideration. The authors need to discuss more the effects of cell transplantation, especially number of cells at one time (between group 1 and group 3) and administration times (between group 1 and group 2 ,and between group 3 and group 4).

We thank you for this comment. To clarify the experimental design, we rewrote section 2.2, providing details about the 3-NP animal models and the methods for the cell injections. As requested, we also improved section 3.4 and the discussion section, providing a detailed discussion about the effects of cell dose and the regimen of cell administration (single or three consecutive doses) in the debate.

  1. From the experimental design, rat brain tissues were collected at 4 days and 30 days after hIDPSCs administration. What is the reason for 4 day and 30 day -points?

We apologize that this information was not evident in the main text. According to scientific literature, the reported MSC lifetime varied between three to seven days after intravenous transplantation (Eggenhofer et al., 2012). However, these data are conflicting (Madec et al., 2009; Pearson et al., 2016) . Therefore, we euthanatized rats four days and 30 days after the cell infusion to verify whether the cells (administrated through an intravenous route) would show a short or long life within the brain.

Following the demonstration that the cells were able to engraft within the cortex and striatum, we analyzed the expression levels of BDNF, DARPP32, and D2R after 30 days (for single-cell administration) and even 90 days after the first cell administration (for the 3 doses of hIDPSCs). The main goal of this analysis was to provide a fast proof of concept. We clarified this point; section 2.4 was rewritten.

Eggenhofer, E. et al. Mesenchymal stem cells are short-lived and do not migrate beyond the lungs after intravenous infusion. Front. Immunol. 3, 1–8 (2012).

Madec, A. M. et al. Mesenchymal stem cells protect NOD mice from diabetes by inducing regulatory T cells. Diabetologia 52, 1391–1399 (2009).

Pearson, J. A., Wong, F. S. & Wen, L. The importance of the non obese diabetic (NOD) mouse model in autoimmune diabetes. J. Autoimmun. 66, 76–88 (2016).

  1. Page5 line 232-234 and Figure 1B : Expression of BDNF in hIDPSCs were shown by immunocytochemistry. However, the method led to this result is not described. Please check this point.

The description of immunocytochemistry was added in section 2.3 (Cell characterization).

  1. Page6 line 257-261 and Figure 2D,2E : Expression of CD73 and CD105 in rat brain were shown by immunofluorescence. However, the methods led to these results are not described. Please check these points.

The method describing the immunofluorescence results was added in section 2.8 (Indirect immunofluorescence).

  1. Page8 line 295: Is scale bars of figure3E 100 µm? Please check it.

The bar scale of figure 3E (which now is Figure 4E) was corrected for 50 μm.

  1. Page8 line 299-304: In this part, figure number and figure legends description do not match. Please check it.

  1. The figure numbers and description in the manuscript were checked and updated since a novel figure demonstrating the striatal damages induced by the 3-NP was added.

  1. Page9 line 320-327: In this part, figure number and figure legends description do not match. Please check it.

The figure numbers and description in the manuscript were checked and updated since a novel figure demonstrating the striatal damages induced by the 3-NP was added.

  1. Page11 line 369: Although it is describe as “1×107cells(G3 and G5)”,G5 is not in Table1. Please check it.

We apologize for this mistake. Line 369 was corrected to (G3). There is no G5 group. This data was corrected in the manuscript

Reviewer 3 Report

In this study, Wenceslau et al investigated about the effectiveness of human mesenchymal cells (hIDPSC) for the therapy of Huntington’s disease (HD) in a rat model. They have found that the hIDPSC improved the pathology of HD. Although the results are promising, some more data are needed to improve the manuscript. I am listing my concern below:

  1. To establish the model and the effectiveness of hIDPSC, the authors need show the basic pathology changes using pathological staining like HE and Nissl. Also, it will be great if the authors show KB staining to show the changes of fibers in the striatum
  2. Functional improvements are important. The authors need to show neurological analysis after the treatment with hiDPSC.
  3. As I understand the authors used several models of treatments, like 1 month or 3 months etc. However, I see that the data of 1 model, which was sacrificed 4 days after treatment. So, it is better to show the study plan as a schematic design showing which model used for what purpose. Also, please describe the results more clearly and in details.
  4. The authors wrote that some transplanted cells looks like pericytes or neurons. This is an important issue. I suggest to check in vivo differentiation potentials by co-localization of immunostaining of cell type marker and vibrant.
  5. The authors did a xenotransplant. So, there will be some immune response. This issue needs to investigate, such as systemic inflammation markers, local inflammatory cells.
  6. The authors need to see how this transplantation improved the pathology, by reducing cell death, improves the circulation or by inducing neurogenesis.

Author Response

In this study, Wenceslau et al investigated about the effectiveness of human mesenchymal cells (hIDPSC) for the therapy of Huntington’s disease (HD) in a rat model. They have found that the hIDPSC improved the pathology of HD. Although the results are promising, some more data are needed to improve the manuscript. I am listing my concern below:

  1. To establish the model and the effectiveness of hIDPSC, the authors need show the basic pathology changes using pathological staining like HE and Nissl. Also, it will be great if the authors show KB staining to show the changes of fibers in the striatum

We thank you for this comment and apologize for not showing the Nissl staining results. As requested, we added the Nissl staining confirming the striatal damages promoted by the 3-NP treatment (Figure 2).

  1. Functional improvements are important. The authors need to show neurological analysis after the treatment with hiDPSC.

We thank you for this comment and agree that functional improvements are essential to the benefits of the treatment with the hIDPSCs. However, we would like to inform you that the results shown in our manuscript are part of a Regulatory Report that had the main goal to demonstrate that the cells could cross the blood-brain barrier and restore the expression of BDNF DARPP32, and D2R. Based on the results presented in our manuscript, we received authorization from the Brazilian Regulatory Agency to perform the Phase I and II clinical trials, which are available on Clinical.Trials.gov (Phase I - https://clinicaltrials.gov/ct2/show/NCT04315987, and Phase II - https://clinicaltrials.gov/ct2/show/NCT03252535). This is because, from the Regulatory perspective, the increase in the expression of these three markers is enough to provide evidence of the therapeutic potential of the hIDPSCs. However, we are developing a novel preclinical test based on the intravenous injection of 1 X 106 cells/Kg (in a single dose) using 3-NP rat models to further investigate the mechanism of action of these cells and analyze the motor and cognitive improvements. This study is under analysis, and it should be published soon.

  1. As I understand the authors used several models of treatments, like 1 month or 3 months etc. However, I see that the data of 1 model, which was sacrificed 4 days after treatment. So, it is better to show the study plan as a schematic design showing which model used for what purpose. Also, please describe the results more clearly and in details.

We thank you for this comment. To clarify our experimental design, we rewritten section 2.4, providing detailed information about the 3-NP animal model, the cell doses used, and the treatment.

  1. The authors wrote that some transplanted cells looks like pericytes or neurons. This is an important issue. I suggest to check in vivo differentiation potentials by co-localization of immunostaining of cell type marker and vibrant.

We appreciated this comment. In fact, we observed that the cells that engraft within the cortex and striatum show a pericyte- and neuron-like morphology, as demonstrated in figure 3. However, as demonstrated by the immunodetection of CD73 and C105, these cells remain expressing MSC hallmarks, providing evidence that the transplanted cells do not differentiate into neurons in vivo. Although we had demonstrated that the hIDPSCs can differentiate into neuron-like cells under in vitro conditions (Kerkis et al., 2006), to date we do not have evidence indicating that these cells differentiate into functional neurons in vivo. Moreover, in the last decade cumulative evidence showed that the therapeutic potential of MSCs is mediated by trophic factors which are naturally produced and secreted in a free form or into extracellular vesicles, as discussed by us in two reviews (Araldi et al., 2020 and Costa et al., 2021).

Araldi et al. Stem cell-derived exossomos as therapeutic approach for neurodegenerative disorders: From biology to biotechnology, Cells, 2020. DOI 10.3390/cellss9122663

Costa et al. Exosomes in tumor Microenvironment: From biology to clinical applications, Cells, 2021, DOI 10.3390/cells10102617

  1. The authors did a xenotransplant. So, there will be some immune response. This issue needs to investigate, such as systemic inflammation markers, local inflammatory cells.

We thank you for this comment and concern. However, we would like to clarify that this study is part of a Regulatory Report, previously shown and accepted by the Brazilian Regulatory Agency as part of the requests needed to approve a clinical trial protocol. Thus, we investigated all criteria proposed by the ISCT to ensure the safety of our cells for clinical purposes. In this sense, using appropriate in vitro models, we showed that the hIDPSCs reduce the expression of IL-2, IL-17, and increase the expression of IL-10, demonstrating the immunomodulatory/anti-inflammatory potential of these cells. These data were presented in a Regulatory Report, which is being prepared for publication. However, due to this analysis, combined with other safety control analyses, the Brazilian Regulatory Agency (ANVISA) approved the phase I and II clinical trials for HD, and COVID-19. These studies are available on the Clinical.Trial.gov:

Phase I clinical trial for HD: https://clinicaltrials.gov/ct2/show/NCT04315987

Phase II clinical trial for HD: https://clinicaltrials.gov/ct2/show/NCT03252535

Phase I/II clinical trial for COVID-19: https://clinicaltrials.gov/ct2/show/NCT04315987

  1. The authors need to see how this transplantation improved the pathology, by reducing cell death, improves the circulation or by inducing neurogenesis.

We thank you for this comment. Please, verify the response to comment 2.

Round 2

Reviewer 1 Report

The authors have answered my comments in a clear and proper way and adapted the manuscript accordingly. I do not have any further comments.
I want only to notify two typos:
"Through" instead of "though" in line 64
"HD" instead of "DH" in line 97

Author Response

We apologize for these typos. They were corrected and the English grammar was reviewed.

Reviewer 3 Report

  1. Xenotransplantation is still an issue that needs to be resolved. I understand that the manuscript is written based on a study that was done for a regulatory report. Since the journal is not a part of that regulatory agency, the rules and regulations do not apply for this paper. Also, the improvement observed by cell transplantation could be the modulation of systemic inflammation, and the local effects of transplanted cells could be minimum. MSC is known to modulate systemic inflammation. Hence, the authors should check the systemic inflammation in 3-NP, untreated and 3-NP+MSC treated rats, and show blood counts, spleen cell profile and histology in the revised manuscript.
  2. Still I do not understand the study design. In my previous comments, I suggested to show the study design as a schematic diagram, where they can show which model was used for which experiment. Also, in the results and figures, they need to mention which model was used to generate that particular result. Then, the readers will understand the meaning of the results. The authors said they prepared many models, but reading the results I am seeing the data of only 1 model. This issue needs to be resolved.

Author Response

Xenotransplantation is still an issue that needs to be resolved. I understand that the manuscript is written based on a study that was done for a regulatory report. Since the journal is not a part of that regulatory agency, the rules and regulations do not apply for this paper. Also, the improvement observed by cell transplantation could be the modulation of systemic inflammation, and the local effects of transplanted cells could be minimum. MSC is known to modulate systemic inflammation. Hence, the authors should check the systemic inflammation in 3-NP, untreated and 3-NP+MSC treated rats, and show blood counts, spleen cell profile and histology in the revised manuscript.

We thank you for this comment. We understand that the journal (Cells) is not part of the regulatory agency and, therefore, the rules and regulations do not apply to this paper. We only informed that the results described in this manuscript were previously presented to the Brazilian Regulatory Agency (ANVISA) in order to demonstrate that our results were enough to provide evidence of the therapeutic potential of human immature dental pulp stem cells (hIDPSCs), allowing the authorization for of the first-in-human clinical trial involving hIDPSCs for the Huntington’s disease treatment. 

 About “Xenotransplantation”: Many experiments have been carried out in which MSCs are transplanted into other organisms of the same or different species. These cells are not rejected because MSCs have shallow levels of MHC class II proteins and lack MHC class I proteins and cannot, therefore, present exogenous antigens to the recipient (host) organism (Augello et al., 2005; Tse et al., 2003; Le Blanc et al., 2003; De Miguel et al., 2012). As a result, they are perceived as endogenous. It is important to realize that the temporary presence of MSCs is not a result of the host immune response since experiments in injured mice with and without functional immune systems yield the same results (Yoo et al., 2013).

In brain injury models, MSC treatment reduces the presence of microglia in the damaged brain and decreases the number of peripheral infiltrating leukocytes at the injured site by increasing anti-inflammatory cytokines (Zhang et al., 2013). In other words, MSCs can be transferred between organisms without eliciting immune rejection by the host, which renders them very good candidates for transplantation, immunosuppression, and immunomodulation (Le Blancet al., 2006).  

Augello A, Tasso R, Negrini SM, Amateis A, Indiveri F, Cancedda R, et al. Bone marrow mesenchymal progenitor cells inhibit lymphocyte proliferation by activation of the programmed death 1 pathway. Eur J Immunol. 2005;35:1482–90.

Tse WT, Pendleton JD, Beyer WM, Egalka MC, Guinan EC. Suppression of allogeneic T-cell proliferation by human marrow stromal cells: implications in transplantation. Transplantation. 2003;75:389–97.

 Le Blanc K, Tammik C, Rosendahl K, Zetterberg E, Ringden O. HLA expression and immunologic properties of differentiated and undifferentiated mesenchymal stem cells. Exp Hematol. 2003;31:890–6.

De Miguel MP, Fuentes-Julian S, Blazquez-Martinez A, Pascual CY, Aller MA, Arias J, et al. Immunosuppressive properties of mesenchymal stem cells: advances and applications. Curr Mol Med. 2012;12:574–91. 10

Yoo SW, Chang DY, Lee HS, Kim GH, Park JS, Ryu BY, et al. Immune following suppression mesenchymal stem cell transplantation in the ischemic brain is mediated by TGF-beta. Neurobiol Dis. 2013;58:249–57.

Zhang R, Liu Y, Yan K, Chen L, Chen XR, Li P, et al. Anti-inflammatory and immunomodulatory mechanisms of mesenchymal stem cell transplantation in experimental traumatic brain injury. J Neuroinflammation. 2013;10:106.

Le Blanc K, Ringden O. Mesenchymal stem cells: properties and role in clinical bone marrow transplantation. Curr Opin Immunol. 2006;18:586–91

About systemic inflammation: We appreciate the comment that the “improvement observed by cell transplantation could be the modulation of systemic inflammation, and the local effects of transplanted cells could be minimum”. However, we do not completely agree with this statement. Although it is recognized that the dental pulp stem cells exhibit a high immunomodulatory potential (Andrukhov et al., 2019; Paganelli et al., 2021; Madhoun et al., 2021), in previous studies, based on RNA-seq analysis, we demonstrated that the hIDPSCs produce and secrete high levels of different neurotrophins within exosomes. These data were described in patent (in analysis in the USA), suggesting that the mechanism of action (MoA) of these cells is based on the transfer of both mRNAs which encode neurotrophins, as well as neurotrophins, including the BDNF, which expression was demonstrated in our manuscript. Moreover, many studies describing the immunomodulatory properties of hIDPSCs were performed using circulating immune cells. In this sense, in an independent study (in preparation to be submitted), we demonstrated that the co-culture of hIDPSCs with activated peripheral blood mononuclear cells (PBMCs) or Jurkat cells are able to reduce the expression levels of interleukin (IL)-2, IL-17 and, increase the levels of IL-10, confirming the anti-inflammatory potential of hIDPSCs. However, Huntington’s disease phenotype is not related to systemic inflammation, but rather to neuroinflammation. In this sense, it is known that neuroinflammation is a consequence of the mitochondrial dysfunction promoted by the mHtt accumulation. Mitochondrial dysfunction increases the reactive oxygen species production, resulting in the microglia activation. Once activated the microglia produce and secrete a plethora of pro-inflammatory cytokines, causing the neuroinflammation (Kerkis et al. 2022). In this regard, although the 3-NP causes the death of striatal neurons, including medium spiny neurons, as demonstrated in our study using Nissl staining, the 3-NP rat model is not appropriate to study the neuroinflammation or the systemic inflammation. Besides this, as previously informed, our study had the main goal to demonstrate that the intravenous injection of hIDPSCs can restore the expression of BDNF, DARPP32, and D2R in 3-NP rats (as emphasized by the title of the manuscript). Additional evidence about the MoA of hIDPSCs are being collected in a novel preclinical study using 3-NP rats, which are being treated with exosomes derived from the conditioned culture medium from hIDPSCs.

 Andrukhov et al. (2019). Immunomodulatory properties of dental tissue-derived mesenchymal stem cells: Implication in disease and tissue regeneration. World J Stem Cells. 11 (9): 604-617

Paganelli et al. (2021). Immunomodulating profile of dental mesenchymal stromal cells: A comprehensive overview. Frontiers in Oral Health. DOI 10.3389/froh.2021.635055

 Madhoun et al. (2021). Dental pulp stem cells derived from adult third molar tooth: A brief review. Frontiers in Cell and Developmental Biology. DOI 10.3389/fcell.2021.717624

 Kerkis et al. (2022). Advances in cellular and cell-free therapy medicinal products for Huntington disease treatment. IntechOpen. DOI: 10.5772/intechopen.102539

Still I do not understand the study design. In my previous comments, I suggested to show the study design as a schematic diagram, where they can show which model was used for which experiment. Also, in the results and figures, they need to mention which model was used to generate that particular result. Then, the readers will understand the meaning of the results. The authors said they prepared many models, but reading the results I am seeing the data of only 1 model. This issue needs to be resolved

As requested, we added a novel figure (Figure 1) schematically describing the study design. We also mentioned at the final of each figure legend the model that was used to generate the results. In order to clarify this point, we also included the analyses that were performed on each end-point in the schematic model of the study design. We hope that this information could resolve the interpretation of the study design, which is also detailed in sections 2, 4, and Table 1.

This manuscript is a resubmission of an earlier submission. The following is a list of the peer review reports and author responses from that submission.

Round 1

Reviewer 1 Report

Wenceslau et al. described the treatment effect of dental pulp stem cells on BDNF, DARPP32, and D2R in a Huntington’s disease rat model.  Dental pulp stem cell is a new cellular resource for therapeutic approaches to neural repair and regeneration. Here the authors provide evidences that dental pulp stem cells administration increased BDNF, DARPP32 and D2R expression levels in the striatum and cortex of a 3-NP- injected rat model.

Many issues decrease the quality and scientific soundness of this manuscript.

  • The authors analyzed the expression levels of BDNF, DARPP32 and D2R using immunohistochemical staining only. Immunohistochemical staining provide a semiquantitative evaluation, which can be affected by fixation, duration, antibody specificity, dilution and so on. The results need to be validated by additional methods, e.g. western blot analysis, ELISA assay. The results of this study are not sufficient for drawing conclusion.
  • The description of data analyses is very poor. There is no information described, how the cell numbers are quantified: using which methods (biased sampling? unbiased sampling?) what is the exact region of interest? in how many animals (animals’ number was only described for total 60, but no information can be found for each group).
  • No sufficient information was presented in figures. In figure 3-5, only representative images for the 3-N + hDPSC were showed, although the later results showed the differences of protein expression levels in different 3-N + hDPSC groups (G1-G4). A representative image of each group is required. Especially, in the figure 5, only images of the striatum present no image of the cortex.
  • No detail description on statistical method. In the figure 6, it is written in the legend “Results of Dunn’s post-hos test”, I assumed that the authors used Two-way-ANOVA, and Dunn’s test comparing each group with 3-NP + placebo group, but no description on the results of two-way ANOVA was found, neither in the main text, nor in the figure legend. It cannot be evaluated if an appropriate statistic method was used.
  • Authors designed experiments using different cell number and single or repeated administration, but only the results of the comparison between administration with different cell number described, the comparison between single and repeated administration will be quite interesting.

Overall the data of this study is not sufficient for drawing the conclusion. Data analyses and data presentation need to be improved.

Author Response

We thank for the comments and suggestions

  • The authors analyzed the expression levels of BDNF, DARPP32 and D2R using immunohistochemical staining only. Immunohistochemical staining provide a semiquantitative evaluation, which can be affected by fixation, duration, antibody specificity, dilution and so on. The results need to be validated by additional methods, e.g. western blot analysis, ELISA assay. The results of this study are not sufficient for drawing conclusion.

We understand the concern about the use of one only method (immunohistochemistry) to demonstrate the therapeutic potential of the human immature dental pulp stem cells (hIDPSCs). However, we would like to emphasize that this study comprises part of the preclinical data of the advanced cellular therapy which is be licensing by the Brazilian’s Regulatory Agency (ANVISA) and, for this purpose, the immunohistochemistry is the only accepted methodology to demonstrate the effects of the treatment within the brain, since other methods, such as Western blot or even ELISA only can demonstrate the specificity of the antibodies, which was extensively demonstrated in previous studies (dissertations and thesis), but do not allow to demonstrate the site of expression, which is crucial for our purpose.

  • The description of data analyses is very poor. There is no information described, how the cell numbers are quantified: using which methods (biased sampling? unbiased sampling?) what is the exact region of interest? in how many animals (animals’ number was only described for total 60, but no information can be found for each group).

In order to provide details about the immunohistochemistry analysis, the following sentence was added in 2.7 section:” A total of three slides (from each group) were dedicated to the quantitative analysis. For this, six field of different brain areas were captured using a binocular Nikon Eclipse light microscope (Nikon, Japan) at the bright field. Images were captured at × 20 magnification using color video camera Nikon CS-R 1(Nikon, Japan) attached to a computer system. Before capturing the images, the light settings were standardized for all imaging sessions. The brain areas analyzed were: cortex, striatum and subventricular zone (for anti-human nucleus antibody), cortex, caudate nucleus and striatum (for anti-BDNF antibody), cortex and striatum (for anti-DARPP32 antibody) and, striatum (for anti-D2R antibody).”

  • No sufficient information was presented in figures. In figure 3-5, only representative images for the 3-N + hDPSC were showed, although the later results showed the differences of protein expression levels in different 3-N + hDPSC groups (G1-G4). A representative image of each group is required. Especially, in the figure 5, only images of the striatum present no image of the cortex.

We apologize for the mistake in relation to Fig. 5. The figure was corrected, showing the D2R immunodetection in both cortex and striatum

  • No detail description on statistical method. In the figure 6, it is written in the legend “Results of Dunn’s post-hos test”, I assumed that the authors used Two-way-ANOVA, and Dunn’s test comparing each group with 3-NP + placebo group, but no description on the results of two-way ANOVA was found, neither in the main text, nor in the figure legend. It cannot be evaluated if an appropriate statistic method was used.

We apologize for this mistake. The legend of the Fig. 5 was modified and the sentence “Statistical analysis performed using ANOVA two-way, followed by the Dunn's post-hoc test, both with significance levels of 5%” was introduced to clarify the statistical method used.

  • Authors designed experiments using different cell number and single or repeated administration, but only the results of the comparison between administration with different cell number described, the comparison between single and repeated administration will be quite interesting.

We agree that this comparison is very important. However, strategically, these data will be provided in a novel manuscript which will may be submitted in next mouth. This novel manuscript comparatively described the therapeutic effect of different cell doses and routes of administration using different a panel of biomarkers and biodistribution assays.

  • Overall, the data of this study is not sufficient for drawing the conclusion. Data analyses and data presentation need to be improved.

In order to provide the missing details, novel sentences were introduced in both Material and Methods and Discussion sections.

Reviewer 2 Report

In this paper, the authors investigated effects of human immature dental pulp stem cells (hIDPSC) on BDNF, DARPP32 and D2R expression in the brain of Huntington’s disease 3-NP rat model.  Their data showed that, administration of hIDPSC could increase the number of BDNF, DARPP32 and D2R positive stained cells in the striatum and cortex in the 3NP-induced rat model. HIDPSC migrated to the striatum, cortex, and subventricular zone using specific markers for human cells four and thirty days after the intravenous administration. This is an interesting study. The authors did well on preparing their manuscript. The data were presented nicely. However, this study has some disadvantages. My comments are as follows:

Main point:

The authors must provide the data showing whether administration of hIDPSC could improve the symptoms of Huntington’s disease in 3NP-induced rat model. Without this fundamental data, the effects of hIDPSC on the expression of BDNF and other molecules don’t have a strong base.

Other points:

Abstract:

  1. The common word of hIDPSC should be SHED. Please check the reference “Miura M, et al. SHED: stem cells from human exfoliated deciduous teeth. Proc Natl Acad Sci U S A. 2003 May 13;100(10):5807-12. doi: 10.1073/pnas.0937635100.”
  2. The sentence “The intravenous route has dosage advantages, enabling the administration to human patients with HD.” should be deleted because it is just a suggestion.
  3. Please add a short conclusion at the end of the abstract.

Introduction:

This part is little bit too long. Please delete some sentences which are not directly related to this study.

Materials and method:

  1. Please provide the description of Statistical analysis. What kind software did the authors use for the post hoc comparisons? Please report whether data were normally distributed or not.

Results

  1. Line 221, “firtly” should be “firstly”.
  2. Figure 1, please indicate the passage, number and culture conditions of hIDPSC cells in the figure legend.
  3. Figure 6, why there are no error bars on the histograms? The post hoc comparisons should be made with the Bonferroni procedure which is more robust to type 1 error than the Dunn's test.
  4. Line 341, 342, *p<0.05, **p<0.01, ***p<0.001 v.s. which group?

Discussion

Please explain why the administration of hIDPSC could increase the expression of BDNF, DARPP32 and D2R in the brain.

Conclusion

The sentence “However, to confirm this statement, clinical exploration needs” should be “However, to confirm this statement, clinical studies are needed”.

Author Response

We thank for the comments and suggestions

1. The authors must provide the data showing whether administration of hIDPSC could improve the symptoms of Huntington’s disease in 3NP-induced rat model. Without this fundamental data, the effects of hIDPSC on the expression of BDNF and other molecules don’t have a strong base

We thank for this comment. We would like to inform that this is the first study describing the preclinical data involving the intravenous injection of human immature dental pulp stem cells (hIDPSCs) for the treatment of Huntington’s disease. Beside this study, we also analyzed behavioral aspects and the mechanism of action of the cells. However, strategically, these results will be shown in a next manuscript. This because, due the limitation of figure and table number, the presentation of all these data in this publication will impact the quality of the results. However, we would like to emphasize that the BDNF is the main neurotrophin downregulated in patients with Huntington’s disease. This because, the disease leads to the progressive loss of medium spiny neurons, affecting the nigrostriatal pathway and, finally, causing the cortical neurodegeneration. As consequence of this process, it is observed a reduction in the expression of DARPP32 (marker of medium spiny neuros), D2R (marker of neuros belonging to the nigrostriatal pathway) and BDNF (neurotrophin produced by the cortico-striatal pathway). Moreover, it is clear that the mutant huntingtin protein (mHtt) impairs the BDNF transport in cortico-striatal, affecting the survival of striatal neurons. However, we demonstrate in Fig. 1 that hIDPSCs overexpress BDNF, providing evidence that the homing of these cells can restore the expression of BDNF, improving the survival of striatal neurons, which was demonstrated by the increased expression of DARPP32 and D2R in the rats that were treated with the cells. In order to clarify this point in the manuscript, we add an introductory paragraph in the discussion: “The aggregated of mutated huntingtin protein (mHtt) verified in patients with Huntington’s disease leads to loss of DARPP32-expressing medium spiny neuros and D2R-expressing neurons of the nigrostriatal pathway. This occurs because the mHtt impairs the BDNF transport in cortico-striatal pathway, affecting the survival of these neurons. Although the 3-NP rat models do not exhibit the aggregations of mHtt, studies already demonstrated that the acid induces selective mitochondrial damages in striatal neurons, particularly in medium spiny neurons, which are able to mimics the pathophysiology of HD.”

2. The common word of hIDPSC should be SHED. Please check the reference “Miura M, et al. SHED: stem cells from human exfoliated deciduous teeth. Proc Natl Acad Sci U S A. 2003 May 13;100(10):5807-12. doi: 10.1073/pnas.0937635100.”

Although our cells are isolated from the dental pulp of children teeth, the human immature dental pulp stem cells obtained by our team are isolated using a novel technology developed by Kerkis et al. (2006) which does not employ any enzymatic treatment. Moreover, Kerkis and Caplan (2012) demonstrated differences in molecular profiler of SHED and hIDPSCs. According to this study, we verified that the SHED overexpress STRO-1, which is poorly or not expressed in hIDPSCs, while the main marker of hIDPSCs in Nestin. Due to this technical difference, our cells cannot classified as SHED.

Kerkis I, Caplan AI. Stem cells in dental pulp of deciduous teeth. Tissue Eng Part B Rev. 2012 Apr;18(2):129-38

3. The sentence “The intravenous route has dosage advantages, enabling the administration to human patients with HD.” should be deleted because it is just a suggestion.

As suggested, the sentence was deleted.

4. Please add a short conclusion at the end of the abstract

A final sentence was introduced in the abstract: “Altogether, these data suggest that the intravenous administration of hIDPSCs can restore the BDNF, DARPP32 and D2R expression, promoting neuroprotection and neurogenesis”

5. This part is little bit too long. Please delete some sentences which are not directly related to this study.

We exclude the sentence “However, complications such as pneumonia, heart disease, and physical injury from falls reduce life expectancy to around 20 years from when symptoms begin” in the first paragraph of introduction. However, the other information cannot be excluded, because they are crucial for the problem presentation.

6. Please provide the description of Statistical analysis. What kind software did the authors use for the post hoc comparisons? Please report whether data were normally distributed or not

As requested, a section describing the statistical analysis were introduced in Material and method section (section 2.8).

7. Line 221, “firtly” should be “firstly”.

We apologize for the orthograph mistake. The word was corrected.

8. Figure 1, please indicate the passage, number and culture conditions of hIDPSC cells in the figure legend.

The passage was informed in figured legend as requested. The culture conditions were informed in section 2.2 (“After thawing hIDPSCs (at passages 3–4) were seeded into culture flasks (150 cm2; Corning, Corning, NY, USA) in Dulbecco’s modified Eagle’s medium (DMEM)/Ham’s F12 (1:1; Invitrogen, Carlsbad, CA, USA), supplemented with 15% fetal bovine serum (FBS; HyClone, Logan, UT, USA), 2 mM glutamine (Gibco, Gaithersburg, MD, USA), 50 mg/ml gentamicin sulfate (Schering-Plough, Whitehouse Station, NJ, USA), and 1% nonessential amino acid (Gibco, Carlsbad, CA, USA). Next, the cells were expanded for two to three more passages. The medium was changed every two days, and the cells were grown until they reach semi-confluence (80–90%). All cell cultures were maintained at 37°C in a 5% CO2 high-humidity atmosphere.”). For the immunodetection of Nesting and BDNF, it was seeded a total of 1 X 103 cells in 1-well chamber slide. This information was introduced in section 2.5.

9. Figure 6, why there are no error bars on the histograms? The post hoc comparisons should be made with the Bonferroni procedure which is more robust to type 1 error than the Dunn's test.

We apologize for the Fig. 6 legend. We would like to clarify that the post-hoc test used was the Bonferroni’s test and not Dunn’s test. This mistake was corrected in both figure legend and in the section  2.8

10. Line 341, 342, *p<0.05, **p<0.01, ***p<0.001 v.s. which group?

In relation to CG3 group (3-NP + placebo). This information was introduced in the Fig. 6 legend.

11. Please explain why the administration of hIDPSC could increase the expression of BDNF, DARPP32 and D2R in the brain.

In order to clarify how the administration of hIDPSCs could increase the expression of these proteins, more details were provided in the discussion.

12. The sentence “However, to confirm this statement, clinical exploration needs” should be “However, to confirm this statement, clinical studies are needed”

The sentence was changed as requested

Round 2

Reviewer 1 Report

The revised manuscript by Wenceslau et al. attempts to address points
raised by previous reviews. Authors present an interesting observation that hIDPSC administration in a 3NP lesion modell restored BDNF-, DARPP32- and D2R expression levels in the striatum and cortex. However, the investigation in this study based on a single method immunohistological staining, which is not appropriate for quantification.  Furthermore, a very small group size (3 slices /group, it is not even clear 3 slices from 3 different animals of from a single animal) crucially decrease the data reliability. Authors argue that immunohistological staining is the only feasible methods due to licensing. I will suggest to publish this data with other parts together to show a complete exploration and convincing analysis.

Minor comments:

In Figure 2. and 3., either authors used wrong size of scale bar or made mistakes in the figure legends. It is clear that b2, b3, c2 and c3 have a different magnification compare to b1, c1, b4 and c4, but it was described in the legend that the scale bars indicate same size. Same issue in Figure 3.  B vs D, F vs H. In many images scale bar is missing.

Figure 2. b1 and b2 are same images, different magnification, c3 doesn’t look like an image of cortex.

Figure 4. Authors used A, B, C, D in both upper panel and lower panel, but these letter in both panels are represented as different groups. Authors also reused same images, it is quite confusing.

Figure 5. If each column doesn’t represent one single value but the average of one group, either standard deviation or standard error should be shown.

Reviewer 2 Report

The authors answered my questions and improved their manuscript. The papar can be accepted.